# Enterococci as Dominant Xylose Utilizing Lactic Acid Bacteria in Eri Silkworm Midgut and the Potential Use of *Enterococcus hirae* as Probiotic for Eri Culture

**DOI:** 10.3390/insects13020136

**Published:** 2022-01-27

**Authors:** Kridsada Unban, Augchararat Klongklaew, Pratthana Kodchasee, Punnita Pamueangmun, Kalidas Shetty, Chartchai Khanongnuch

**Affiliations:** 1Division of Biotechnology, School of Agro-Industry, Faculty of Agro-Industry, Chiang Mai University, Mueang, Chiang Mai 50100, Thailand; kridsada_u@cmu.ac.th; 2Interdisciplinary Program in Biotechnology, The Graduate School, Chiang Mai University, Mueang, Chiang Mai 50200, Thailand; augchararat.taey@gmail.com (A.K.); pratthana_kod@cmu.ac.th (P.K.); pia_pannita39@hotmail.com (P.P.); 3Department of Plant Sciences, Global Institute of Food Security and International Agriculture (GIFSIA), North Dakota State University, Fargo, ND 58108, USA; kalidas.shetty@ndsu.edu; 4Research Center for Multidisciplinary Approaches to Miang, Science and Technology Research Institute Chiang Mai University, Mueang, Chiang Mai 50200, Thailand

**Keywords:** Eri silkworm, pentose-utilizing microbe, lactic acid bacteria, probiotic, Eri culture

## Abstract

**Simple Summary:**

This study focused on isolation and identification of xylose utilizing lactic acid bacteria from the midgut of Eri silkworm to understand their characteristics such as tannin tolerance, production of cellulolytic enzymes, and antimicrobial activity against insect pathogenic bacteria. The *Enterococcus* was found as the dominant genus among xylose utilizing lactic acid bacteria. Within this genus, *Enterococcus hirae* SX2 showed the potential to be used as a probiotic in Eri silkworm culture due to its tannin tolerance and antimicrobial activity against insect pathogenic bacteria. The trial experiment for applying live *E. hirae* SX2 supplemented to castor leaves in Eri silkworm rearing showed a positive effect for improving larval weight and survival. These findings led to the development of a new probiotic for Eri culture and also could be the experimental model for screening of the potential probiotic from mulberry silkworm (*Bombyx mori*).

**Abstract:**

A total of 51 pentose utilizing lactic acid bacteria (LAB) were isolated from acid-forming bacteria in the midgut of healthy mature Eri silkworm using de Man, Rogosa and Sharpe (MRS) agar containing 10 g/L xylose (MRS-xylose) as the carbon source supplemented with 0.04% (*w*/*v*) bromocresol purple. Further analysis of 16S rRNA gene sequences revealed the highest prevalence of up to 35 enterococci isolates, which included 20 isolates of *Enterococcus mundtii*, followed by *Entercoccus faecalis* (eight isolates), *Weissella cibaria* (four isolates), *Enterococcus hirae* (two isolates), *Enterococcus lactis* (one isolate), and *Enterococcus faecium* (one isolate). All 51 LAB isolates showed positive growth on MRS containing a range of polysaccharides as the sole carbon source. All isolates were able to grow and form clear zones on MRS supplemented with 1 g/L xylose, while *E. faecalis* SC1, *E. faecalis* SCT2, and *E. hirae* SX2 showed tannin tolerance ability up to 5 g/L. Moreover, five isolates showed antimicrobial activity against Eri silkworm pathogens, including *Bacillus cereus*, *Staphylococcus aureus*, and *Proteus vulgaris*, with *E. hirae* SX2 having the highest inhibitory effect. Supplementation of live *E. hirae* SX2 on castor leaves significantly improved the weight and reduced the silkworm mortality when compared with the control group (*p* < 0.05). This cocci LAB can be considered as the new probiotic for Eri culture. Additionally, this finding presented the perspective of non-mulberry silkworm that could also be used as the model for further applying to new trends of the sericulture industry.

## 1. Introduction

Eri silk is non-mulberry silk that is becoming more popular due to improvements in relevant silkworm domestication and multivoltine nature of silk. Eri silk is made from Eri silkworm (*Samia ricini*), which is believed to be originally found in northeast India, primarily in Assam and Meghalaya, and then distributed to other Asian countries [1]. The cultivation of Eri silkworm has benefits and relevance in rural-based on-farm and off-farm activity, which has been recognized as a beneficial approach for socio-economic development, particularly in some Asian developing countries [2]. Furthermore, the textile products from Eri silk are promoted as green products due to their natural process and eco-friendly properties [3]. Since the normal feed of Eri silkworm is a variety of plant leaves that are structurally composed of cellulose, hemicelluloses such as xylan and mannan, pectin, lignin, and the small quantities of protein and fat [4], the gut of this Lepidopteran has a major role in the digestion of leaf components for nutrition. The microbial community in the insect gut has been extensively studied and discussed regarding their symbiont characteristics and role as beneficial gut microbiota where their metabolically relevant enzymes are required for digestion of the substrates and release of nutritionally important compounds [5]. The bacterial community in the midgut of this Lepidopteran insect has also been investigated for the production of some digestive enzymes required for the digestion of leaf components [6]. Several reports concluded that gut microbiota influences the insects in several ways, such as supporting nutrient digestion and detoxification [7], improving innate immunity [8], providing nutrients and growth-promoting metabolites [9], and protecting against infectious pathogens and parasites [10]. Though several studies reported on the gut microbiota linked to insect metabolism and growth development, the relationship between the gut microbiota community and their possible role for advancing applications in Eri silk production are not fully understood.

Microbial biodiversity is now increasingly considered essential for the metabolism of higher eukaryotic systems, and therefore understanding relevant microbial community in Eri silkworm gut could advance these microbial resources in Eri silkworm production as a source of beneficial probiotic lactic acid bacteria. This has been specifically targeted as a source of optically pure L- or D-form producing microbes resulting in biological process for the production of optically pure L- or D-lactic acid, particularly from pentose substrates composed in lignocellulose that is required for the manufacturing of bioplastics such as polylactic acid (PLA) [11,12]. Besides the utilization as the biological catalyst for the production of lactic acid, some lactic acid bacteria (LAB) such as *Lactobacillus* sp. and *Bifidobacterium* sp. have been investigated and applied as probiotics for providing health benefits both for humans and animals. The effect of probiotic bacteria on the improvement of growth and quality of silkworm has also been reported [13,14,15,16]. However, most previous studies have investigated the mulberry silkworm (*Bombyx mori*), and none has focused on Eri silkworm. Since the important characteristic of LAB probiotics is the generation of antimicrobial metabolites for countering pathogenic or undesirable microbes in the host gastrointestinal tract [17], the antimicrobial activity against insect pathogens in response to LAB is also considered as an important characteristic of probiotics for silkworm cultivation. In addition, the application of live probiotics is more eco-friendly for the management of silkworm diseases rather than the use of antibiotics, which is widely used in the sericulture industry [18].

Therefore, based on the above background and rationale in this study, the isolation of xylose utilizing LAB and their molecular identification aligned with their antimicrobial activities against insect pathogenic bacteria were investigated. Further investigation of applying selected xylose utilizing LAB as probiotic in Eri culture was also advanced.

## 2. Materials and Methods

### 2.1. Isolation and Screening of Xylose Utilizing LAB

Ten mature Eri silkworm larvae (5th instar day 5) were collected from a local farm in Sankampang district, Chiang Mai province, Thailand. The larvae were rinsed twice with sterile water and surface sterilized in ethanol (70%, *v*/*v*) for 60 s, followed by a final rinsing in sterile distilled water before dissection. Then, the worm’s gut tissue was aseptically dissected and immediately transferred to a sterile microcentrifuge tube containing sterile 0.85% (*w*/*v*) NaCl solution in a ratio of 1:10. A 10-fold serial dilution was carried out by pipetting 0.5 mL of homogenized sample into 4.5 mL of 0.85% (*w*/*v*) NaCl solution. The mixture was then mixed well, and 100 μL of each dilution was transferred to nutrient agar (NA) containing 10 g/L glucose and 0.04% (*w*/*v*) bromocresol purple using the spread plate technique. After 24 h incubation at 37 °C, the acid-forming bacterial isolates observed from the yellow clear zone around their colonies were then transferred into NA containing 10 g/L xylose and 0.04% (*w*/*v*) bromocresol purple. The viable bacterial cells were enumerated as colony-forming units (CFU) per gram of sample.

All acid-forming bacterial colonies capable of growth on NA xylose were transferred into modified de Man, Rogosa and Sharpe (MRS) agar containing 10 g/L beef extract, 10 g/L peptone, 5 g/L yeast extract, 5 g/L sodium acetate, 2 g/L K_2_HPO_4_, 2 g/L tri-ammonium citrate, 0.2 g/L MgSO_4_, 0.2 g/L MnSO_4_, 0.1% (*v*/*v*) Tween80, and 10 g/L xylose supplemented with 0.04% (*w*/*v*) bromocresol purple and then incubated at 37 °C for 24 h. The colonies that exhibited a yellow zone around their respective colony were presumptively considered to be LAB and were individually streaked on MRS-xylose agar plate to obtain single colonies. The cultures of pure isolates were stored at −80 °C in MRS broth mixed with 25% (*v*/*v*) glycerol.

### 2.2. Identification of LAB by 16S rDNA Sequence Analysis

The molecular identification of LAB isolates was carried out using the method described by Unban et al. [19] with some modification. Briefly, the bacterial cells were collected from 10 mL of an overnight bacterial culture by centrifugation at 10,000× *g* for 10 min, and the genomic DNA was extracted by using the Wizard Genomic DNA purification kit (Promega Corp., Madison, WI, USA), according to the manufacturer’s instructions. The 16S rRNA gene was amplified by using genomic DNA as a template with a standard polymerase chain reaction (PCR). Forward primers 27F, 5′-AGAGTTTGATCMTGGCTCAG-3′, and reverse primers 1525R, 5′-AAGGAGGTGWTCCARCC-3′, were used for 16S rRNA gene amplification. The amplification cycles were initially performed at 94 °C for 5 min of denaturation, followed by 30 cycles of denaturation at 94 °C for 20 s, annealing at 55 °C for 20 s, extension at 72 °C for 1 min 30 s, and additional final extension at 72 °C for 5 min. All PCR reactions were performed using Phusion High-Fidelity PCR Master Mix (New England Biolabs, MA, USA). The PCR products were purified and sequenced by a sequencing service provider (1st BASE Laboratory Company, Singapore). The comparison of 16S rRNA gene sequences was performed with the genetic database from National Center for Biotechnology Information (NCBI) GenBank and EzBioCloud databases. The multiple sequence alignment was performed using BioEdit 7.0 software tool and MEGA 4.0 [20]. The phylogenetic tree was constructed by the neighbor-joining methods with 1000 replications bootstrap analysis. All 16S rRNA gene sequences generated in this study have been deposited in the NCBI GenBank database under accession number MZ127632–MZ127643 and OM090176–OM090214.

### 2.3. Growth and Acid Formation on Polysaccharides

The selected strains of xylose utilizing LAB from Eri silkworm midgut were investigated for their abilities to utilize the selected polysaccharides as the sole carbon source. Briefly, individual LAB isolate was spiked on MRS agar containing 0.5% (*w*/*v*) of single carbon source of individual polysaccharides including starch (Fisher Scientific, Loughborough, Leicestershire, UK), pectin (Fisher Scientific, Loughborough, Leicestershire, UK), beechwood xylan (Megazyme International, Bray, Co. Wicklow, Ireland), locust bean gum (Sigma Aldrich, St. Louis, MO, USA) or carboxymethyl cellulose (Sigma Aldrich, St. Louis, MO, USA) supplemented with bromocresol purple (0.04%, *w*/*v*). After incubation at 37 °C for 24 h, the bacterial growth and the yellow halo-formed surrounding colonies were observed. Further duplicated experiment was also carried out, but the bromocresol purple was replaced with trypan blue (0.01%, *w*/*v*) for detection of extracellular activities of amylase, pectinase, xylanase, β-mannanase, and cellulase, respectively. The clear zone formed surrounding colonies were observed after incubation at 37 °C for 24 h.

### 2.4. Tannin-Tolerance Test

The ability of LAB isolates to tolerate tannins was evaluated on MRS-xylose agar supplemented with tannin. To prepare tannin solution, 10 g of tannic acid (LOBA Chemie, Mumbai, India) was dissolved in 100 mL of 0.1 M sodium phosphate buffer (pH 7.0) and sterilized by filtering through a 0.2 μm filter cartridge (Millipore, Billerica, MA, USA). The tannin solution was aseptically added into the sterilized MRS-xylose agar to achieve the final concentrations of 1 and 5 g/L. A single colony of each LAB isolate was picked up and spiked on the supplemented MRS-xylose agar. Growth of bacterial colony and the yellow clear zone formation of the isolates were observed after incubation at 37 °C for 24 h.

### 2.5. Antimicrobial Activity against Insect Pathogens

The antimicrobial activity against insect pathogenic bacteria was investigated by independent cultivation of all selected LAB isolates in MRS-xylose broth at 37 °C for 24 h. Then, cell-free culture supernatants (CFCS) were collected by centrifugation at 10,000× *g* for 15 min. The CFCS were then neutralized to pH 6.5 with 5 N NaOH and filtered through a 0.22 µm filter cartridge (Millipore, Billerica, MA, USA). The antimicrobial activity was investigated by the disc-agar diffusion method following the method of Fehlberg et al. [21]. Briefly, 100 μL of the overnight grown pathogen including *B. cereus* TISTR 747, *S. aureus* TISTR 746, and *P. vulgaris* TISTR 100 cultures (10^6^–10^7^ CFU/mL) was spread onto NA agar plate, and paper discs (diameter 6 mm, Macherey-Nagel, Duren, Germany) were placed on each plate. Then, 50 µL of CFCS from LAB isolates were transferred into paper discs. The diameter of the inhibition zone against pathogenic bacteria was measured after incubation at 37 °C for 24 h.

### 2.6. Effect of E. hirae SX2 Supplementation on Eri Silkworm Growth

A pure culture of *E. hirae* SX2 was inoculated in MRS-xylose broth and incubated at 37 °C under static conditions for 24 h. The bacterial cells were collected by centrifugation at 8000× *g*, 4 °C for 10 min and cell pellets was washed twice with sterile 0.85% (*w*/*v*) NaCl solution, and the cell pellets were resuspended in NaCl solution and adjusted to an optical density (OD600) of around 0.5 corresponding to approximately 10^8^ CFU/mL. The freshly picked castor leaves were cleaned by excess volume of tap water and finally washed with sterile water. The cleaned leaves targeted for feeding were then sprayed with a cell suspension of *E. hirae* SX2 on both sides of leaves and shade dried before feeding. The 1st instar larvae of Eri silkworm were divided into two groups for the treatment, with each group consisting of 50 larvae; one group was reared on castor leaves served as control, while the other was fed on probiotics treated leaves. Three replications were maintained for each treatment. The treatment was given the first feed on the first day of 1st instar larvae up to 5th instar stage (3 weeks) under laboratory conditions at 25 ± 1 °C, the humidity of about 75 ± 5%, and photoperiod of 16 h of light and 8 h of dark. The treatment was fed 2–3 times a day, and the unfed leaves were removed from the trays daily. The growth of larva was monitored (10 larvae per treatment were selected randomly) by recording the larva weight once a week. At the end of rearing (3 weeks), 10 larvae per treatment were randomly selected as the representative of each treatment for dissection, and the midgut content were collected and mixed well for use as the representative of each group for determination of total bacterial count on NA agar supplemented with 10 g/L glucose and 0.04% (*w*/*v*) bromocresol purple as an indicator. The viable cell number of acid-forming bacteria was also enumerated and observed from the yellow clear zone formed surrounding the colony. The viable cell number of *E. hirae* SX2 was enumerated by spread plating on MRS-xylose agar supplemented with 0.04% bromocresol purple.

### 2.7. Statistical Analysis

The collected data were analyzed using the statistical program SPSS/PC version 17.0 (SPSS Inc. Chicago, IL, USA). The data were analyzed for achieving both normality and homoscedasticity, and the statistical significance of the differences among the treatments was evaluated by one-way ANOVA followed by Tukey’s multiple range test. Values not sharing a common letter are significantly different from each other at *p* < 0.05.

## 3. Results and Discussion

### 3.1. Xylose Utilizing LAB Isolation

The healthy mature fifth instar Eri silkworm larvae cultivated at the local farm in Sankampang district, Chiang Mai, Thailand (Figure 1a) was used as the source of midgut content. The numbers of acid-forming bacteria that grew on different kinds of media are presented in Figure 1b. The total viable bacterial population of the Eri silkworm midgut observed on NA glucose were found to be 3.95 × 10^7^ CFU/g of larval gut, while acid-forming bacteria were found at a high number of 3.76 × 10^7^ CFU/g, which represented 95.4% of the total number of bacteria. Thereafter, 1000 isolates of these acid-forming bacteria were randomly replicated into NA-xylose and incubated at 37 °C for 24 h. Only 608 isolates (60.8%) showed the ability to grow and produce acid around the colonies. Of all isolates that were able to grow and produce acid in NA-xylose, only 51 isolates (10.2%) showed an acid-forming capability in the selective MRS-xylose agar and were presumptively identified as LAB. These results indicated that even xylose utilizing bacteria were found in significant numbers among acid-forming bacteria capable of growth in NA, but LAB was a minor component representing only 10.2% of the total number when tested on the LAB selecting medium such as MRS agar. These results were in agreement with the previous study, which suggested that the main population of culturable bacteria in Eri silkworm gut were non-LAB, including *Enterobacter* sp., *Pseudomonas* sp., *Citrobacter* sp., and *Bacillus* sp. [22]. This might be due to the pH condition of the insect gut, which is reported to be alkaline [10], which may limit the growth and number of LAB.

### 3.2. Xylose Utilizing LAB Identification and Phylogenetic Analysis

The morphological study of 51 LAB isolates found that 47 isolates were Gram-positive with cocci shape, while the last four isolates were Gram-positive with irregular rod shape. All the xylose utilizing LAB isolates were identified using the sequence analysis of the 16S rRNA gene. The sequence was aligned with the related type strains, and the construction of phylogenetic trees was created and compared with the sequences of their closest relatives (Figure 2a). The full-length sequence of 16S rRNA gene of LAB isolates shared similarity to genera *Enterococcus* and *Weissella* with higher than 99%. Of 51 isolates, 35 were identified to be *Enterococcus mundtii*, which represented 68.6% of total isolates, followed by *Entercoccus faecalis* (15.7%), *Weissella cibaria* (7.8%), *Enterococcus hirae* (3.9%), *Enterococcus lactis* (2%), and *Enterococcus faecium* (2%) (Figure 2b). *Enterococcus* and *Lactobacillus* have been reported to be the predominant LAB genera in mulberry silkworm midgut, where the relative abundance varied depending on silkworm species and their physiological activities [23]. In another study, the predominant gut microbiota from the healthy silkworm larvae were *Delfitia* sp., *Ralstonia* sp., *Enterococcus* sp., *Staphylococcus* sp., and *Pelomonas* sp.; nevertheless, the observation of relative abundance change depended on the larval stage [24]. According to Chen et al. [25], approximately 10^7^ CFU of bacteria were found in the whole gut of each sample. The identification using 16S rDNA sequence found that *Enterococcus* (62.1%) and *Clostridium* (35.4%) were the two main genera in each sample. Furthermore, the effect of bacteria including *Lactobacillus* spp., *Bifidobacterium*, *B. licheniformis*, and *B. niabensis* reflected in the improvement of growth and productivity characteristics of silkworm *Bombyx mori* L. [15,26,27]. Several studies recently have reported microbial abundance and diversity of Lepidopteran insect guts [22,23,28,29], and some xylose utilizing bacteria in Lepidoptera were also reported [16,30].

### 3.3. Growth and Acid Formation on Polysaccharide Substrates

Silkworm does not code for cellulolytic genes; therefore, some of the cellulolytic-degrading enzymes might be produced by gut microbiota [31]. All selected xylose utilizing LAB do not show the extracellular polysaccharide degrading enzymes production on MRS agar supplemented with various polysaccharide substrates. However, all selected LAB showed positive growth, but the acid formation was found in varying levels when cultured on MRS agar supplemented with individual polysaccharides as the sole carbon source (data not shown). This means that these LAB may possess membrane-bound enzymes with the capability to convert these polysaccharides to free simple sugars and assimilate them into their metabolism via the lactic acid fermentation pathway. The results of this study revealed that some xylose utilizing LAB isolated in this study did not utilize various polysaccharide substrates by directly secreting the extracellular polysaccharide degrading enzymes, which is different from previous research where *Enterococcus* sp. was the host gut microbe with the capability of cellulolytic enzyme activity [14,32]. This evidence leads to a further suggestion that the main role of polysaccharides degradation in Eri silkworm may be undertaken by other non-LAB microbes such as the well-recognized polysaccharides degrading bacteria *Bacillus* spp., which is also reported to be the dominant bacterial population in the midgut of *Bombyx mori* L. silkworm [14,15].

### 3.4. Tannin-Tolerance Test

The results from these experiments found that all 51 LAB selected isolates showed tolerance against tannins supplemented in MRS-xylose agar at the final concentration of 1 g/L tannin with variation observed based on the extent of yellow clear zones around their colonies (Figure 3). The clear zones formed by 51 isolates on MRS-xylose with 1 g/L of tannin were found in the range of 4–15 mm, which was comparable to a lesser degree in size to their clear zone on the MRS-xylose agar without tannin supplementation. Interestingly, only three isolates (*E. faecalis* SC1, *E. faecalis* SCT2, and *E. hirae* SX2) were able to grow and form yellow clear zones on MRS-xylose supplemented with 5 g/L of tannin (Figure 3c), whereas the other isolates could not grow under this condition. Phenolic compounds of plants are secondary metabolites, including such as flavonoids, tannins, and phenolic acids, which exhibit many biological activities associated with plant growth and development [33]. Tannin is the polyphenolic compound that is commonly found in plant leaves, and this phenolic compound has been recognized for its antimicrobial activity effect as a result of binding either with the cellular enzymes in the cytoplasm or cell surface receptor proteins in the cell wall [34]. Thus, it seems likely that a tannin-rich environment provides a selective condition to the LAB that can survive in tannin-rich conditions. The capability of growth of *E. faecalis* SC1, *E. faecalis* SCT2, and *E. hirae* SX2 indicates the special beneficial characteristic for active survival of these LAB in the tannin-rich condition in Eri silkworm gut.

### 3.5. Antimicrobial Activity

One of the common benefits provided by probiotics is the ability to combat infection of pathogenic bacteria. In this study, *B. cereus*, *S. aureus*, and *P. vulgaris* were used as Gram-positive and Gram-negative pathogenic indicators, and the result is presented in Table 1. Overall, all 51 isolates exhibited inhibitory effects against at least one of the tested pathogens. Five isolates with the most excellent antagonistic activity against all three tested insect pathogenic bacteria were *E. faecalis* SCT2, *E. hirae* SX2, *E. mundtii* UDF1, *E. mundtii* UDFX1, and *E. mundtii* UDG3. *P. vulgaris* was found to be the most sensitive pathogenic strain based on the CFCS inhibitory effect produced by LAB isolates. The highest inhibitory effect was observed in *E. hirae* SX2 against all tested pathogenic bacteria (Figure 4).

Some previous reports confirmed that the silkworm gut is colonized with many bacteria [22,35]. *Streptococcus* sp. was found to be the most abundant pathogen in *Bombyx mori* larvae, whereas LAB from genus *Pediococcus*, *Leuconostoc*, and *Lactobacillus* did not cause any infection to silkworm [36]. However, the beneficial effect on silkworm host or interaction among the different microflora and the precise mechanism of action is unclear. Even though microbial supplementation as probiotics is commercially used for humans, ruminants, poultry, and fisheries, there are only few options of a probiotic formulation designed for silkworms [36]. Meanwhile, applying the commercial probiotic formulation was found to improve the larval body weight, effective rate of rearing, cocoon weight, pupal weight, shell ratio, and silk productivity [37,38,39,40]. Since synthetic antibiotics are widely applied in sericulture for disease control and some previous studies showed the antibiotic potential of LAB from silkworm against a range of both Gram-positive and Gram-negative pathogenic bacteria, considering the use of LAB isolated from silkworm gut with the antimicrobial activity has been proposed to be an alternative way for ecofriendly management of silkworm diseases [18]. Furthermore, some enterococci strains showed their ability to produce several bacteriocins [41], which confirm a competitive advantage toward the pathogenic microbes, and some *Enterococcus* spp. have been reported to be used as probiotics, such as *E. faecium*, *E. faecalis*, *E. lactis*, *E. hirae*, and *E. durans* [15,42,43]. However, the potential of enterococci used as probiotics supplement in Eri silkworm has not yet been reported. The dominant strains of xylose utilizing *Enterococcus* spp. isolated from the healthy silkworm of this study, and particularly *E. hirae* SX2 has the potential to contribute beneficial effects to the silkworm, and it is expected to be used as probiotic for Eri silkworm culture.

### 3.6. Effect of E. hirae SX2 Supplementation on Eri Silkworm Growth

The effects of presumptive probiotic *E. hirae* SX2 on the growth of Eri silkworm are shown in Table 2. The results revealed that the larvae fed on probiotics treated castor leaves exhibited a significant difference in weight when compared to the naturally fed control group. After 3 weeks of cultivation (fifth instar), larval weight of Eri silkworm reared with castor leaves supplemented with *E. hirae* SX2 were 4.53 ± 0.07 g, whereas larval weights of the control group were 4.20 ± 0.04 g. The results showed that larval weight of Eri silkworm reared with probiotic showed a significant increase of 7.85% (*p* < 0.05) when compared to the control leaves. In addition, the mortality rate of 3.3% was observed from the Eri silkworm reared with castor leaves supplemented with *E. hirae* SX2, which was 9.3% lower than that of the control. The culturable bacteria of Eri silkworm midgut reared on probiotic after 3 weeks of cultivation were enumerated (Figure 5). The results revealed that probiotic supplementation significantly affected total viable cells in the Eri silkworm gut. The high numbers of total viable cells and acid-forming bacteria in probiotic-treated larvae were detected when compared with untreated. Furthermore, the viable cells of LAB significantly increased when compared to the control treatment of castor leaves. These results indicated that probiotic *E. hirae* SX2 might be able to survive and grow in larval gut and could provide some benefits to the host larvae quality leading to the increase of the larval weight.

The probiotic microbial flora activity in the host gut might increase the efficiency of digestion and food assimilation, potentially leading to increase protein synthesis and assimilation in the host [44]. Moreover, probiotics have the ability to produce some vitamins and rapidly degrade the digestible compounds, potentially leading to nutritional assimilation. Based on the theory that beneficial microbial ecology is essential for eukaryotic metabolism, insects need gut microbial flora to provide a diversity of enzymes required for digestion of the nutritional components, which can contribute to the release of amino acids, fermentable sugars, and other molecules, which are beneficial to the growth of the insect [45]. Furthermore, the effect of probiotic bacteria including *Lactobacillus* spp., *Bifidobacterium*, *B. licheniformis*, and *B. niabensis* exhibited improvement in growth and quality characteristics of silkworm *Bombyx mori* L. [15,16,26,27]. Even some enterococci strains produce several bacteriocins, which have been reported with reference to the use of probiotics as mentioned previously, but there are no reports of the use of enterococci as probiotics supplement in Eri silkworm. The present study is the first report that indicates the potent beneficial effects of probiotic *Enterococcus* sp. on the Eri silkworm larval growth. However, more details concerning the influencing factors affecting the efficacy of probiotics, such as the probiotic dosage and probiotic formulation, are essential for advancing practical uses in Eri culture.

## 4. Conclusions

In this study, the xylose utilizing LAB from midgut of Eri silkworm larvae were isolated and characterized for evaluating probiotic characteristics such as tannin tolerance, production of cellulolytic enzymes, and antimicrobial activity against insect pathogenic bacteria, which potentially benefit Eri silkworm. Furthermore, among 51 LAB isolated from Eri silkworm, *E. hirae* SX2 was selected as the potential probiotic LAB based on beneficial characteristics such as tolerance to 0.5% (*w*/*v*) tannin and antimicrobial activity against insect pathogenic bacteria. The initial evaluation for targeting application of *E. hirae* SX2 in Eri silkworm rearing showed the beneficial and positive effect for improving larval quality, and this enterococci LAB is proposed as a new probiotic for Eri silk culture. Moreover, these findings will help to improve Eri silkworm economic traits and could also be used as the model in the further development of novel probiotics for silkworm (*B. mori*) or sericulture research.

## Figures and Tables

**Figure 1 insects-13-00136-f001:**
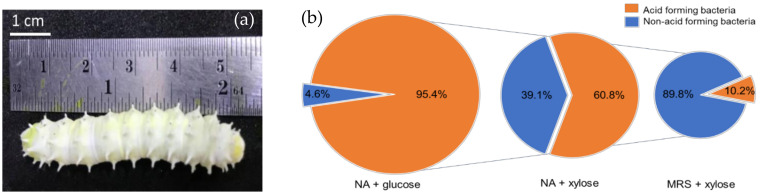
The mature Eri silkworm larvae cultivated at local farm in Chiang Mai, Thailand (**a**) and the pie diagram presenting the percentage of acid-forming bacteria from Eri silkworm midgut found in different agar medium (**b**).

**Figure 2 insects-13-00136-f002:**
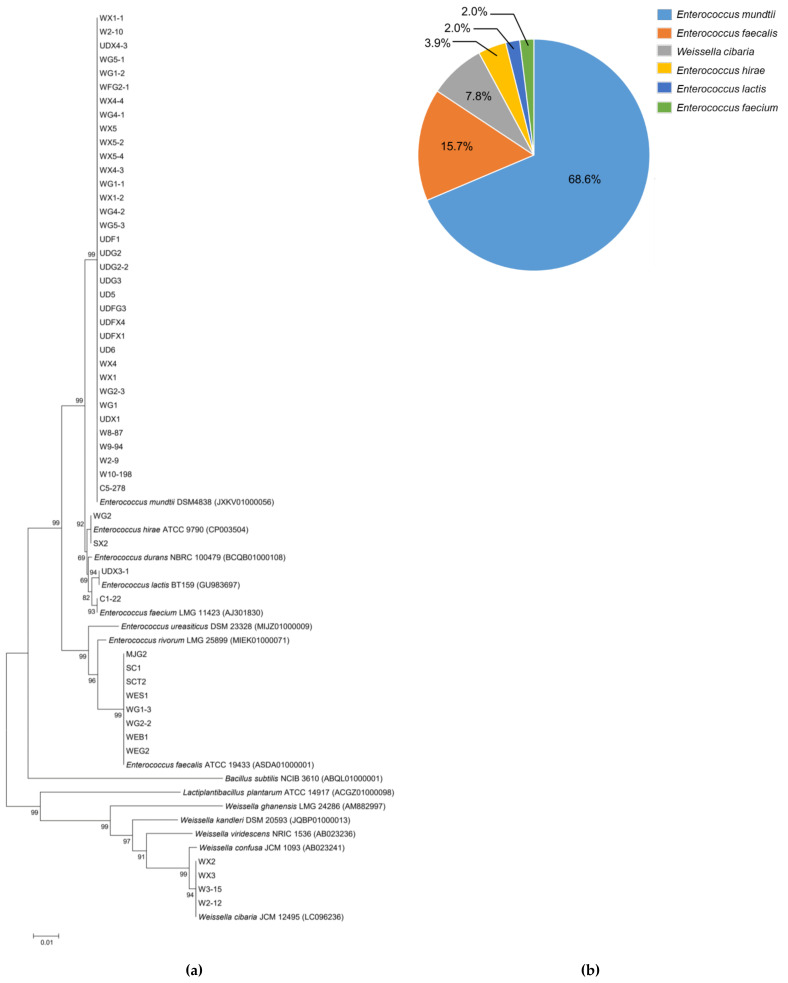
Phylogenetic tree based on the 16S rRNA gene sequence analysis (**a**) and the proportion percentage of the identified species (**b**) of 51 selected LAB from Eri silkworm midgut.

**Figure 3 insects-13-00136-f003:**
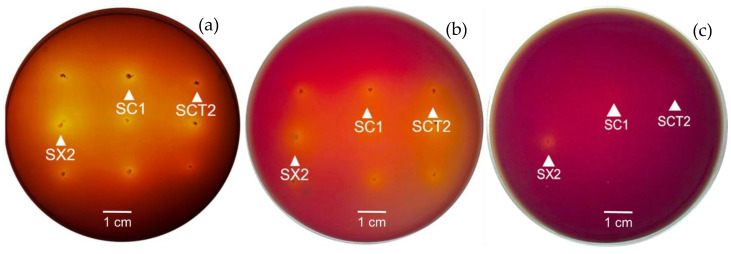
Growth and clear zone formation of xylose-utilizing LAB, *E. faecalis* SC1, *E. faecalis* SCT2, and *E. hirae* SX2, on MRS-xylose agar (**a**), MRS-xylose agar supplemented with 1 g/L tannin (**b**), and 5 g/L tannin (**c**), when cultivated at 37 °C for 24 h.

**Figure 4 insects-13-00136-f004:**
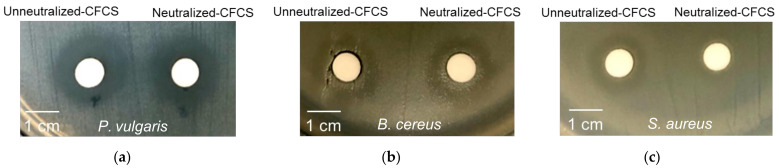
Antimicrobial activity of cell-free culture supernatant (CFCS) of *E. hirae* SX2 cultivated in MRS xylose against *Proteus vulgaris* (**a**), *Bacillus cereus* (**b**), and *Streptococcus aureus* (**c**).

**Figure 5 insects-13-00136-f005:**
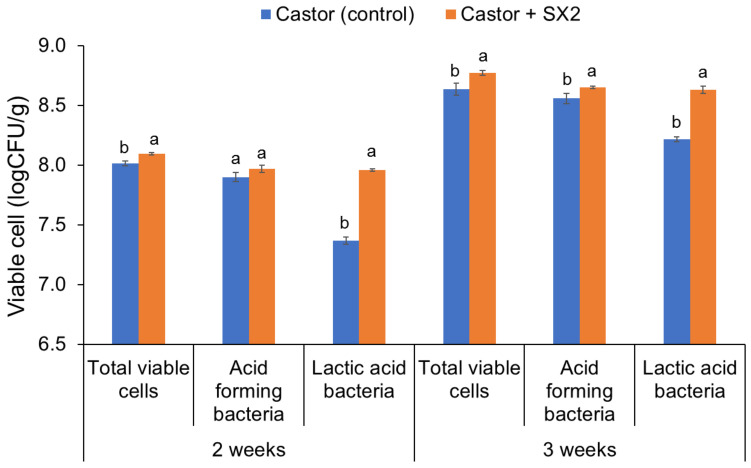
Total viable cells, acid-forming cells, and lactic acid-forming cells of probiotic-supplemented Eri silkworm midgut. Different small letters indicate significant differences in the viable cell (logCFU/g) in midgut between Eri silkworm reared with castor leaves and *E. hirae* SX2 treated castor (*p* < 0.05).

**Table 1 insects-13-00136-t001:** Antimicrobial activity of LAB isolates against some pathogenic bacteria.

Isolates	Inhibition Zones against Pathogenic Bacteria (mm)					
	*Proteus vulgaris*		*Bacillus cereus*		*Streptococcus aureus*	
	Unneutralized CFCS	Neutralized CFCS	Unneutralized CFCS	Neutralized CFCS	Unneutralized CFCS	Neutralized CFCS
C1-22	++	++	+	+	ND	ND
C5-278	++	++	+	+	ND	ND
MJG2	++	++	+	+	ND	ND
SC1	++	++	+	+	ND	ND
SCT2	+++	+++	++	+	++	+
SX2	+++	+++	++	++	++	+
UD5	+++	+++	+	+	ND	ND
UD6	++	++	+	+	ND	ND
UDF1	+++	+++	++	+	+	ND
UDFG3	+	+	+	+	ND	ND
UDFX1	+++	+++	++	+	+	ND
UDFX4	++	++	+	+	ND	ND
UDG2	++	++	+	+	ND	ND
UDG2-2	++	++	+	+	ND	ND
UDG3	+++	+++	++	+	+	ND
UDX1	++	++	+	+	ND	ND
UDX3-1	++	++	+	+	ND	ND
UDX4-3	++	++	+	+	ND	ND
W10-198	++	++	+	+	ND	ND
W2-10	++	++	+	+	ND	ND
W2-12	++	++	+	+	ND	ND
W2-9	++	++	+	+	ND	ND
W3-15	++	++	+	+	ND	ND
W8-87	++	++	+	+	ND	ND
W9-94	++	++	+	+	ND	ND
WEB1	++	++	+	+	ND	ND
WEG2	++	++	+	+	ND	ND
WES1	+++	+++	+	+	ND	ND
WFG2-1	+	+	+	+	ND	ND
WG1	++	++	+	+	ND	ND
WG1-1	++	++	+	+	ND	ND
WG1-2	+	+	+	+	ND	ND
WG1-3	+	+	+	+	ND	ND
WG2	++	++	+	+	ND	ND
WG2-2	++	++	+	+	ND	ND
WG2-3	+	+	+	+	ND	ND
WG4-1	+	+	+	+	ND	ND
WG4-2	++	++	+	+	ND	ND
WG5-1	+	+	+	+	ND	ND
WG5-3	+	+	+	+	ND	ND
WX1	+	+	+	+	ND	ND
WX1-1	+	+	+	+	ND	ND
WX1-2	++	++	+	+	ND	ND
WX2	+++	+++	++	+	ND	ND
WX3	+++	+++	++	+	ND	ND
WX4	+++	+++	+	+	ND	ND
WX4-3	+++	+++	+	+	ND	ND
WX4-4	++	++	+	+	ND	ND
WX5	++	++	+	+	ND	ND
WX5-2	++	++	+	+	ND	ND
WX5-4	++	++	+	+	ND	ND

Note: clear zone around disc; +: 1–3 mm; ++: 3–5 mm; +++: >5 mm; ND: no inhibition zone was detected.

**Table 2 insects-13-00136-t002:** Weight (g) of Eri silkworm reared with castor leaves and *E. hirae* SX2 supplemented castor leaves during 3 weeks of cultivation.

Treatments	0 Week (1st Instar Larva)	1 Week	2 Weeks	3 Weeks
Castor	0.03 ± 0.04	0.48 ± 0.03 ^a^	1.53 ± 0.04 ^b^	4.20 ± 0.04 ^b^
Castor + SX2	0.02 ± 0.02	0.47 ± 0.01 ^a^	1.69 ± 0.09 ^a^	4.53 ± 0.07 ^a^

Note: means in columns with different superscripts are statistically different at *p* < 0.05.

## Data Availability

The data presented in this study are available in article.

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
