# Peer review of "Enterococci as Dominant Xylose Utilizing Lactic Acid Bacteria in Eri Silkworm Midgut and the Potential Use of Enterococcus hirae as Probiotic for Eri Culture"

_insects, 2022, doi:10.3390/insects13020136_

Round 1
Reviewer 1 Report
This work has interesting aims and could entail potential applied interest. I have however a number of issues that the authors need to address which I will outline as follows.
Line 201 Statistics: “the statistical significance of the differences among the treatments was evaluated by one-way ANOVA followed by Tukey’s multiple range test.”
Can the authors provide the normality test results for data distribution and the results of the homoskedasticity for means /medians, to ascertain that ANOVA was a legitimate approach and that data would not call for non-parametric estimations?
Line 188: “The treatment was fed 2-3 times a day and the unfed leave were removed from 189 the trays daily.”
How does one know which larvae are unfed?
Authors used 37°C as temperature for bacterial isolation, including total culturable cells. But as insects are not warm-blooded, why did they choose such value that is normally pertinent when isolating enteric bacteria from mammals ? 37°C is also a challenging temperature that may even prevent the growth of some bacteria that could dwell into the silkworm gut.
Are the sequences of the 16S gene used to identify those isolates deposited in a public database as commonly needed ? Please provide the codes or the FASTA files to enable inspection and judgement of the attributions.
Line 176 Paragraph 2.6. “Effect of E. hirae SX2 Supplementation on Eri Silkworm Growth. A pure culture of E. hirae SX2 was inoculated in MRS-xylose broth and incubated at 37°C under static condition for 24 h. The bacterial cells were collected by centrifugation at 8,000´g, 4°C for 10 min and cell pellets was washed twice with sterile 0.85% (w/v) NaCl solution and the cell pellets were resuspended in NaCl solution to obtain a concentration of 10^8 CFU/mL.”
How can one resuspend a pellet and achieve a concentration of 10^8 CFU/mL ? CFU are Colony Forming Units, therefore, in order to know their concentration, one must plate dilutions and incubate for days until colonies could be counted. Moreover, this procedure will yield a number that has infinitesimally low probability to be a perfectly ‘round’ number as 10^8 CFU/mL. In other words, a concentration as precise as this one could be achieved by appropriately diluting whatever suspension, but only after knowing the actual number of CFU/ml. However, in the case of what the authors describe, since the CFU/ml information requires at least one day of time, how could they then ‘go back in time’ to that suspension and act accordingly? Even in the case one would wait by storing the suspension in the cold, the actual numbers would be altered (upwards) by some residual completion of cell divisions and (downwards) by an unknown rate of viability/culturability loss during the storage. Besides, this is not what the authors affirm to have done. The question therefore is: how can they explain what they have declared to have done?
As regards the experiment whose results are presented in Fig. 5 This enumeration of different colony phenotypes is technically unclear. Data are obtained by plating and counting. As values up to billions of colony forming units per gram are scored, the counts must have involved multiple serial dilutions and counting plates such as 10-8 in order to have countable (non overlapping/coalescent) colonies, i.e. in numbers around <300 per plate. But, as the medium is non- selective the two sub-populations (acid forming and Lactic acid bacteria) involve respectively the visualization of a yellow halo as fraction of the total cells, and the transfer on MRS medium of the latter (acid forming) to inspect for haloes on MRS. Given this technical necessity, since the numbers of the different phenotypes could be up to half a log lower, can the authors actually describe what they did to obtain those numbers avoiding the fact that the number of colonies of the prevailing phenotypes could overwhelm/mask the minority ones ?
In the initial experiment with mature larvae of the 5th instar the LAB positive cases were 10% of those 60.8 % of the acid forming bacteria, which in turn were 94,5% of the total. Therefore the LAB should be about 4,8 of the total.
How came that, for the experiment whose results are shown in Fig. 5, at 5 weeks the total viable cells were around 8,7 log CFU/g and LAB were around 8,4 log CFU/g which is about 50% of the total viable cells (instead of less than 5% as resulting from the experiment in Fig. 1 ? The larvae are declared to be of equivalent age (fifth instar) as in line 346 authors report: “After 3 weeks of cultivation (5th instar), larval weight…”
In addition to this, at 2 weeks (7.4 logCFU/g vs. 8 logCFU/g ) LAB were therefore about 1/4 of the total, how do the authors explain these dynamics?
Moreover, in the SX-supplemented larvae, data show that the LAB on total community become almost 100%. This is so different from the 4.8% precent found in mature larvae of the same age that it would imply that the added strain would really compete within the resident community and prevent its expected development. But notwithstanding this, Fig. 5 shows that In both control and SFx- treated, there is an increase of the whole community; those data are CFU/g, and there appears to be a nearly 1-log increment in the bacteria per gram of gut in a week (which is not due to the mere larval growth as data are not per larva but per g of gut). Is there any rational interpretation for these fluctuations and can the data of Fig.1 be reconciled with those of Fig.5 ?
In essence, I feel that the authors need to provide a thorough addressing to all these points in order for their work to be revaluated.
Reviewer 2 Report
In your paper, you isolated the xylose utilizing LAB in the gut of Eri silkworm and evaluating probiotic characteristics potentially benefitting Eri silkworm. And E. hirae SX2 was selected as the potential probiotic LAB with its tolerance to tannin and antimicrobial activity. It may play an important role in future Eri culture. It is a good work but I have some suggestions and comments.
3.2. Xylose Utilizing LAB Identification and Phylogenetic Analysis, add references on xylose utilizing bacteria in Lepidoptera
How much (the concentration) tannin in the leaf you feed on Eri silkworm?
Line 17, I think it is better to delete the word “isolated”.
Line 27, delete the word “among”.
Line 38, “the highest inhibitory effect was observed in response to E. hirae SX2” can amend to “E. hirae SX2 showed the highest inhibitory effect”.
Line 79, I think “pentose utilizing” should be changed to “pentose-utilizing”.
Line 116, there's an extra parenthesis
Line 256, you can change your sentence to “Silkworm does not code for cellulolytic gene, therefore some of the cellulolytic-degrading enzymes might be produced by gut microbiota”.
Line 257, your sentence “The investigation for……with various polysaccharide substrates” is a little wordy, you can write “All of the selected xylose utilizing LABs do not show the extracellular polysaccharide degrading enzymes production based on MRS agar supplemented with various polysaccharide substrates”.
Line 388, “from midgut of silkworm” should be changed to “from midgut of Eri silkworm”.
From line 316 to line 342, I think this paragraph is redundant for your result.
In Fig. 3, you should add a scale bar.
In your paper, sometimes you use “LAB” and sometimes you use “LABs” please stick to the format.
You can simplify the language in your paper.
For discussion, several highly related papers in the filed should be discussed: Symbiont-Derived Antimicrobials Contribute to the Control of the Lepidopteran Gut Microbiota;
Gut bacterial and fungal communities of the domesticated silkworm (Bombyx mori) and wild mulberry-feeding relatives. And some others.
In your method, you use mature 5th instar Eri silkworm, and I think you should tell us the exactly which day insects have been. I think the gut microorganism of 5th instar larvae can vary a lot in different days. And do you use insects growing on the same day?
In your method 2.6., you use three replications for recording the larva weight, please clearly indicate the number of silkworms you used for weighting?
Proteus vulgaris, Bacillus cereus and Streptococcus aureus are the common pathogens for Eri silkworm?
Reviewer 3 Report
The manuscript provides an interesting contribution to the knowledge of a part of the gut microbiota of the Eri silkworm which is useful in digestion of leaf components of the diet. The Authors, using culture-dependent techniques, selected and isolated from the gut of Eri silkworm larvae bacteria that produce acid from the fermentation of xylose. The molecular identification showed that they were mainly enterococci. Through a functional characterization, an interesting strain of E. hirae was identified. Its use as supplementation to castor leaves fed to larvae of Eri silkworm gave a significant, albeit minimal, beneficial effects with an increase of the larval weight.
Even if the research design is appropriate, the methods well described and the results clearly presented, some minor parts of the manuscript need to be improved and corrected. In particular, a paragraph of the introduction in my opinion needs to be revised because it is not very relevant. A list of specific comments is attached.

Round 2
Reviewer 1 Report
The manuscript still needs a round of English language editing.
